# Effects of Nitrogen Addition on Soil Microbial Functional Diversity and Extracellular Enzyme Activities in Greenhouse Cucumber Cultivation

Zhen Wang [1,2,3,4], Shuang Wang [1,2,3,4], Ting Bian [1,2,3,4], Qiaobo Song [1,2,3,4], Guorui Wu [1,2,3,4], Muhammad Awais [1,2,3,4], Yufeng Liu [1,2,3,4], Hongdan Fu [1,2,3,4,*] and Zhouping Sun [1,2,3,4,*]

1    College of Horticulture, Shenyang Agricultural University, Shenyang 110866, China
2    Key Laboratory of Protected Horticulture of the Education Ministry and Liaoning Province, Shenyang 110866, China
3    National & Local Joint Engineering Research Center of Northern Horticultural Facilities Design & Application Technology, Shenyang 110866, China
4    Collaborative Innovation Center of Protected Vegetable Surround Bohai Gulf Region, Shenyang 110866, China
*    Correspondence: fuhongdan@syau.edu.cn (H.F.); suner116@126.com (Z.S.)

**Abstract:** Greenhouses, commonly used for vegetable production, are experiencing large nitrogen (N) inputs in North China, which leads to soil acidification, increases soil N availability, and affects microbial community structure and composition. However, it remains unclear how N enrichment influences soil microbial functional activities in this region. In this study, we conducted a two-year pot experiment in a greenhouse to evaluate the effects of four different rates of N addition (0, 334, 668, and 1002 kg N ha$^{-1}$ year$^{-1}$) on cucumber soil properties, extracellular enzyme activities, and community level physiological profiles (CLPP). We found that high-N addition (1002 kg N ha$^{-1}$) caused a massive accumulation of inorganic nitrogen and soil acidification, which was not beneficial to soil microbial activities. The color development (AWCD) values for the metabolism of microbial carbon sources and the activities of soil extracellular enzymes also showed a significant decrease in high N(N3) treatment. Additionally, the activity of leucine aminopeptidase (LAP) and polyphenol oxidase (PPO) of N3 decreased by 36% and 50% compared to the N0 and could be a good predictor for microbial functional diversity and microbial biomass carbon (MBC). Structural equation modeling (SEM) confirmed that the reduction of microbial functional diversity is mainly coregulated by the decline of soil pH and the change of cucumber BGB (belowground biomass) resulting from soil C and N imbalance. Overall, excessive N-fertilizer amendment can be more dangerous to microbial community functional diversity, especially for carbohydrate utilization which adversely affects cucumber yield in current intensive management.

**Keywords:** nitrogen fertilization; soil acidification; community level physiological profiles (CLPP); extracellular enzyme activity; microbial functional characteristics; structural equation model





## 1. Introduction

During the past few decades, the input of nitrogen through fertilizers and fossil fuel burning by human activities has increased fourfold and has had a great impact on the terrestrial biosphere [1,2]. As the most fundamental nutrient for crop growth, nitrogen (N) with reasonable addition in an N-limited ecosystem can promote plant biomass and community diversity [3,4]. However, due to the pursuit of economic benefits, the phenomenon of soil acidification caused by overusing nitrogen fertilizers has been particularly serious in intensive agriculture cropping systems, resulting in the deterioration of soil fertility and in yield reductions [5–7].

Microbes play a crucial role in nutrient cycling and material transformation of terrestrial ecosystems [8]. A global meta-analysis showed N addition had a significant influence

on microbial biomass and activity and was generally accompanied by a decline in pH [9]. As a fundamental soil property, soil pH not only affects other soil physicochemical properties but also regulates the soil-microbial-community structure, such as the contents of actinomycetes, bacteria, and fungi based on phospholipid fatty acids (PLFAs) [10,11]. Furthermore, N addition could also regulate soil C/N ratio by enhancing N availability, which affects the acquisition of C, N by microorganisms, and plant C allocation belowground, in turn altering soil microbial activity and community composition [12–14]. Nonetheless, there is not enough data on how soil pH and N availability could affect soil microbial communities, and their relative contribution to microbe variation under N addition remains unclear.

Changes in the soil microbial biomass and community tend to shift their functional activities [15]. Extracellular enzymes (including hydrolases and oxidases) are actively involved in litter decomposition and soil nutrient cycling, which are closely related to microbial function activity [16,17]. N-induced soil acidification has different effects on the soil extracellular enzyme activity because enzyme activity is predominantly controlled by soil pH, which affects enzyme kinetics through changes in substrate binding and stability [18]. Generally, glycosidase and acid phosphatase (ACP) activities are enhanced after low-nitrogen-level addition [16,19]. Excessive N addition has significantly reduced, increased, or had no apparent impacts on the activities of β-1,4-N-acetyl-glucosaminidase activity (NAG) and oxidase in different types of ecosystems [20–22]. Compared with extracellular enzymes, the Biolog Eco-plates method is widely used to correct microbial metabolic patterns, is considered essential to predict soil quality and can describe the microbial functional process at the community level [23,24]. Kumar et al. [25] believed that the decrease in soil pH, the alteration of the C/N ratio, and the increase in litter caused by nitrogen input was the main reason for the variation in the microbial community-level physiological profile. On a low-nitrogen-level grassland, nitrogen fertilization can alleviate soil nitrogen limitation and promote microbial metabolic activity [26], while high nitrogen often has inhibitory effects on microbial carbon source metabolism and functional diversity indices [27–29]. In contrast, Cui et al. [30] reported that high-nitrogen-deposition promoted the Shannon index of an evergreen broad-leaved forest. From the above investigations, we can infer that both extracellular enzymes and microbial metabolic patterns under N addition were comprehensively affected by ecosystem, location-specific, and initial nitrogen levels, which makes our understanding of their response mechanisms more uncertain.

As one of the most common vegetable crops in Chinese greenhouses, cucumber (*Cucumis sativus* L.) accounts for approximately 77% of global production. Due to its high economic importance, planting cucumbers under excess N fertilizer has become an extremely universal phenomenon and a substantial threat to cucumber producers. It is reported that the annual nitrogen fertilizer application rate of greenhouses has exceeded 2000 kg ha$^{-1}$ which is several times higher than that of other cultivation systems, leading to increased soil acidity and nutrient excessive accumulation [31,32]. In recent years, most studies have focused on microbial community structures and the composition of field crops. However, there are relatively few investigations on the effects of nitrogen-fertilizer addition on the soil microbial function activities of cucumber growing systems with high fertility requirements. Thus, we carried out a greenhouse pot experiment with four nitrogen addition levels combined with soil-plant properties, extracellular enzymes, and function characteristics analysis, the primary goals were to (i) explore the response of soil extracellular enzyme activities to N addition; (ii) evaluate the influence of N addition on the microbial community level physiological profiles; (iii) explore whether pH was the main driving factor affecting the alteration of microbial functional diversity and soil extracellular enzyme activities following N addition under greenhouse cucumber cultivation. This evidence provides a basis for maintaining soil ecosystem functions and developing sustainable cucumber-intensive management.

## 2. Materials and Methods

### 2.1. Study Site and Experimental Design

This experiment was executed at a greenhouse located at the experimental site of the Shenyang Agricultural University of Liaoning Province, China (41°48′ N, 123°25′ E). Based on the USDA (United States Department of Agriculture) soil taxonomy, the soils were classified as brown loam. Soils were collected from the open field behind the greenhouse and then sieved through a 1 cm screen to remove stone and plant residues. The N addition experiment was initiated in August 2018. First, the soils were mixed homogeneously with chicken manure compost which was added at a rate of 6.7 g kg$^{-1}$ soil. Then four N-addition-rate treatments with three replicates were designed: a control without N addition (N0), 334 kg N ha$^{-1}$ year$^{-1}$ (N1), 668 kg N ha$^{-1}$ year$^{-1}$ (N2), and 1002 kg N ha$^{-1}$ year$^{-1}$ (N3). Twelve 27 cm wide and 30 cm high polyethylene plastic pots were filled with these soil mixtures. Each pot was filled with 14 kg of soil. We used urea as nitrogen fertilizer. Additionally, potassium sulfate and dipotassium phosphate were added to all the treatment pots at the rates of 166 kg P$_2$O$_5$ ha$^{-1}$ year$^{-1}$ and 234 kg K$_2$O ha$^{-1}$ year$^{-1}$ to satisfy cucumber nutrient demands, which are calculated according to the actual production level of local farmers. The chemical fertilizer was applied three times with water during the cucumber growth period. "Jin You NO.30", a widely grown cucumber breed in this area was cultured in the pots which were randomly arranged and placed in the greenhouse. There were two growth seasons each year for the continuous cucumber potting cropping method.

### 2.2. Soil Sampling and Chemical Analysis

After two years (four crop stubbles) of N addition, we used a five-point sampling method to collect a mixed soil sample from each pot on 6 November 2020. Fresh soils were passed through a 2 mm sieve to remove stones, pebbles, and coarse roots. The soil samples were divided into two parts, one was sent to the laboratory for storage at 4 °C for the extracellular enzyme and CLPP analysis, and the other was dried at room temperature before the determination of chemical properties. Additionally, cucumbers were harvested as fresh vegetables and biomass. Soil pH (deionized water: soil, 2.5:1) was measured using a Thunder Magnetic pHS-25 pH Meter (INESA, Shanghai, China). Soil electrical conductivity (EC) (deionized water: soil, 5:1) was determined by a Thunder Magnetic DDS-307A EC Meter (INESA, Shanghai, China). Soil organic carbon (SOC) was determined by the H$_2$SO$_4$-K$_2$Cr$_2$O$_7$ titrimetric method [33]. Total nitrogen (TN) was determined using an automatic Kjeldahl distillation–titration method [34]. The soil C/N ratio was calculated based on the SOC and TN. The soil nitrate (NO$_3^-$-N) and ammonium (NH$_4^+$-N) were extracted with 0.5 M K$_2$SO$_4$ and then quantified using a flow-injection autoanalyzer (Skalar San$^{++}$ CFA, Erkelenz, Germany) [35]. Total phosphorus (TP) and total potassium (TK) were determined via the H$_2$SO$_4$-HClO$_4$ digestion method [36]. The available phosphorus (AP) was determined using the NaHCO$_3$ leaching molybdenum antimony colorimetric technique [37]. Soil-available potassium (AK) was extracted using 1 mol l$^{-1}$ NH$_4$OAc solution and then analyzed using a flame photometer (iCE3000, Thermo Fisher Scientific, Waltham, MA, USA) [38]. The microbial biomass carbon (MBC) was carried out with the fumigation extraction method with 0.5 M K$_2$SO$_4$ and determined with a total organic C/N analyzer (Multi N/C 3100/HT1300, Analytik Jena AG) [39].

### 2.3. Soil Extracellular-Enzyme-Activity Assays

Five hydrolytic enzymes involving β-1,4-glucosidase (BG) and cellobiohydrolase (CB) for the C cycle, β-1,4-N-acetyl-glucosaminidase (NAG), leucine aminopeptidase (LAP) for the N cycle, acid phosphatase (ACP) for the P cycle, as well as two oxidative enzymes including polyphenol oxidase (PPO) and peroxidase (PER) were the extracellular enzymes we investigated in this study. Enzyme assays were performed according to the protocol described in previous studies by German et al. [40]. In brief, fresh soil (1.5 g) was added to 125 mL of 50 mM sodium acetate buffer (pH = 6.0). Soil slurries and substrate were contained in 96-well microplates and incubated at 25 °C for 2.5 h (hydrolytic enzymes) and

4 h (oxidative enzymes). The quantity of fluorescence or absorbance was determined with a microplate reader (Biotek Synergy 2, Winooski, VT, USA) at 360 nm excitation and 460 nm emission for hydrolytic enzymes and read at 450 nm for oxidative enzymes (Table 1).

**Table 1.** Extracellular enzyme assayed in cucumber soil, their enzyme commission number, and the corresponding substrate.

| Enzyme | Abbreviation | EC Number | Substrate |
|---|---|---|---|
| β-1,4-glucosidase | BG | 3.2.1.21 | 4-MUB-β-D-glucoside |
| Cellobiohydrolase | CB | 3.2.1.91 | 4-MUB-β-D-cellobioside |
| Leucine aminopeptidase | LAP | 3.4.11.1 | Leucine-7-amino-4-methylcoumarin |
| β-1,4-N-acetyl-glucosaminidase | NAG | 3.1.6.1 | 4-MUB-N-acetyl-B-D-glucosaminide |
| Acid phosphatase | ACP | 3.1.3.2 | 4-MUB-phosphate |
| Peroxidase | PER | 1.11.1.7 | L-DOPA |
| Polyphenol oxidase | PPO | 1.10.3.2 | L-DOPA |

*2.4. Soil Community Level Physiological Profile (CLPP) Analysis*

Soil microbial metabolism activity was indicated by the average well-color development (AWCD). Soil microbial functional diversity was evaluated with the Shannon and McIntosh indices. Specifically, the Shannon and McIntosh indices are affected by the richness and evenness of microbial species, respectively [25,41]:

$$AWCD = \sum (C_i - R)/N, \tag{1}$$

$$Shannon = -\sum (P_i \times \ln P) \text{ and } P_i = (C_i - R)/\sum (C_i - R), \tag{2}$$

$$McIntosh = \sqrt{\sum (C_i - R)^2} \tag{3}$$

where $C_i$ represents the absorbance in the microplate well, involving the ith C substrate; R represents the absorbance in the control well, involving sterile water; N represents the total quantity of C substrates included in a specific C group.

*2.5. Statistical Analysis*

The SPSS Software 22.0 (SPSS Inc., Chicago, IL, USA) was adopted to conduct statistical analyses. Effects of nitrogen addition on soil biotic and abiotic parameters were estimated by one-way analysis of variance (ANOVA). The test of least significant difference (LSD) was employed and the difference of $p < 0.05$ was considered significant. The variations in soil microbial metabolic characteristics and extracellular enzyme activities among the N treatments were investigated using the principal component analysis (PCA) combined with PERMANOVA. Redundancy analysis (RDA) and Pearson's correlation analysis were used to evaluate the relationships between microbial community diversity and enzyme activities. In addition, the links among the function of the microbial community, soil enzyme activities, and edaphic properties were determined by Mantel tests. The structural equation modeling (SEM) was conducted in the AMOS 21.0 (SPSS Inc., Chicago, IL, USA) to explore the causal relationships among soil properties, enzyme activities, and microbial C source utilization.

**3. Results**

*3.1. Soil and Plant Properties*

The levels of $NH_4^+$-N and soil electrical conductivity (EC) increased significantly with an increase in nitrogen addition rate, whereas soil pH decreased significantly. Under N2 and N3 treatment, MBC content was significantly lower than that of the control (N0) ($p < 0.05$; Table 2). The content of $NO_3^-$-N, available potassium (AK), total nitrogen (TN), and total phosphorus (TP) were markedly higher at medium-high levels of the nutrient amendment (N2 and N3) compared with a low level of N amendment (N1) and the control (N0). Soil organic carbon (SOC) content in the N1 and N2 treatments was significantly

higher than N0. However, the content of soil-available phosphorus (AP) only exhibited a significant increase in the N2 treatment, and the C/N ratio was significantly lower ($p < 0.05$) in the N3 treatment compared with the N0 treatment. Cucumber yield was significantly higher in N1 compared to N0 after 2 years of N addition and then gradually lowered by 66% from N1 to N3.

**Table 2.** Soil properties, microbial biomass, and cucumber yield in different N addition treatments.

| Environmental Attributes | N0 | N1 | N2 | N3 |
|---|---|---|---|---|
| pH | $7.01 \pm 0.11$ a | $6.38 \pm 0.34$ b | $5.83 \pm 0.09$ c | $5.49 \pm 0.11$ c |
| EC (ms cm$^{-1}$) | $0.33 \pm 0.11$ c | $0.57 \pm 0.12$ bc | $0.7 \pm 0.3$ b | $1.45 \pm 0.11$ a |
| NH$_4^+$-N (mg kg$^{-1}$) | $1.98 \pm 0.57$ c | $5.51 \pm 2.06$ c | $34.72 \pm 4.07$ b | $62.25 \pm 1.83$ a |
| NO$_3^-$-N (mg kg$^{-1}$) | $2.57 \pm 0.61$ b | $29.67 \pm 13.06$ b | $177.94 \pm 66.87$ a | $199.57 \pm 16.09$ a |
| TN (g kg$^{-1}$) | $1.37 \pm 0.1$ c | $1.7 \pm 0.03$ b | $2.01 \pm 0.22$ a | $2.03 \pm 0.18$ a |
| AP (mg kg$^{-1}$) | $236.18 \pm 14.54$ b | $225.48 \pm 7.17$ b | $304.21 \pm 8.19$ a | $240.38 \pm 0.62$ b |
| AK (mg kg$^{-1}$) | $721.05 \pm 90.91$ b | $839 \pm 109.99$ ab | $1031.69 \pm 182.27$ a | $1001.66 \pm 83.9$ a |
| TP (g kg$^{-1}$) | $1.09 \pm 0.12$ b | $1.24 \pm 0.02$ b | $1.45 \pm 0.05$ a | $1.46 \pm 0.17$ a |
| TK (g kg$^{-1}$) | $25.67 \pm 1.7$ a | $23.53 \pm 0.52$ a | $22.93 \pm 1.36$ a | $24.96 \pm 3.5$ a |
| SOC (g kg$^{-1}$) | $24.21 \pm 1.4$ b | $28.56 \pm 2.6$ a | $29.39 \pm 1.8$ a | $26.8 \pm 2.4$ ab |
| MBC (mg kg$^{-1}$) | $396.8 \pm 26.03$ a | $402.06 \pm 7.07$ a | $241.52 \pm 13.4$ b | $121.08 \pm 10.59$ c |
| Soil C/N | $17.7 \pm 2.01$ a | $16.79 \pm 1.47$ ab | $14.74 \pm 2.16$ ab | $13.24 \pm 1.61$ b |
| Yield (kg plant$^{-1}$) | $0.66 \pm 0.06$ b | $1.16 \pm 0.38$ a | $0.76 \pm 0.04$ ab | $0.39 \pm 0.2$ b |

Note: EC, electrical conductivity; SOC, soil organic carbon; TN, total nitrogen; TP, total phosphorus; TK, total potassium; NO$_3^-$-N, nitrate nitrogen; NH$_4^+$-N, ammonium nitrogen; AP, available phosphorus; AK, available potassium; Soil C/N, the ratio of SOC and TN; MBC, microbial biomass carbon; Yield, cucumber yield; All data are presented as mean $\pm$ SD ($n = 3$). Different letters in the same edaphic factor indicate significant ($p < 0.05$) differences between all treatments. N0, N1, N2, and N3 treatments represent the applied N rates of 0, 334, 668, and 1002 kg ha$^{-2}$ y$^{-1}$, respectively.

### 3.2. Soil Enzyme Activities

N addition had significant effects on oxidase and hydrolase activities ($p < 0.05$; Figure 1). Polyphenol oxidase activity increased in the N1 treatment and then gradually decreased with the increasing of N addition rate (from N1 to N3) compared to the control (N0) ($p < 0.05$; Figure 1G). However, peroxidase activity increased under medium treatment and significantly decreased under high-N addition (N3) ($p < 0.05$; Figure 1F). All hydrolases except acid phosphatase showed a similar trend (Figure 1 A–E). Compared to the N0 treatment, the N2 treatment significantly enhanced β-1,4-glucosidase (BG) and β-1,4-N-acetyl-glucosaminidase (NAG) activity by 61.23% and 210.57%, and N3 treatment significantly suppressed cellobiohydrolase (CB) and leucine aminopeptidase (LAP) activity by 50.37% and 36.21%, respectively. Acid phosphatase activity was significantly higher ($p < 0.05$) than the control in all the N-addition treatments, but there was no significant difference among them. The PCA result demonstrated the differences between C-, N-, and P-cycling enzymes following different N addition treatments. The first two principal coordinates explain 41.4% and 28.0% of the total enzyme activities variation. The horizontal axis separates enzyme activities by N addition. N0, N1, and N2 treatments are distributed along the right side of the Y axis, whereas the N3 treatment is on the left side. A statistical test showed that the addition of different N levels had a significant effect on enzyme activity (Figure 2A, PERMANOVA test, $p = 0.001$).

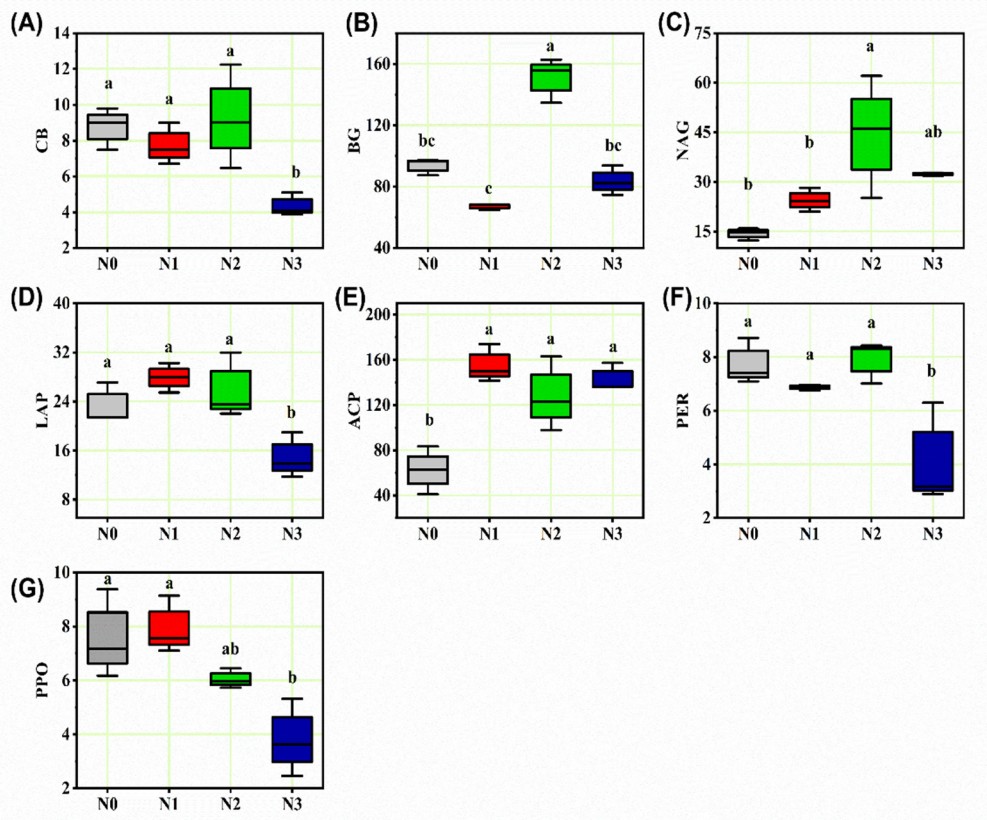

**Figure 1.** Activities of soil β-1,4-glucosidase ((**A**), BG), cellobiohydrolase ((**B**), CB), β-1,4-N-acetyl-glucosaminidase ((**C**), NAG), leucine aminopeptidase ((**D**), LAP), acid phosphatase ((**E**), ACP), peroxidase ((**F**), PER) and polyphenol oxidase ((**G**), PPO) in different N-rate treatments. Different lowercase letters on the boxplot indicate significant differences ($p < 0.05$) among treatments.

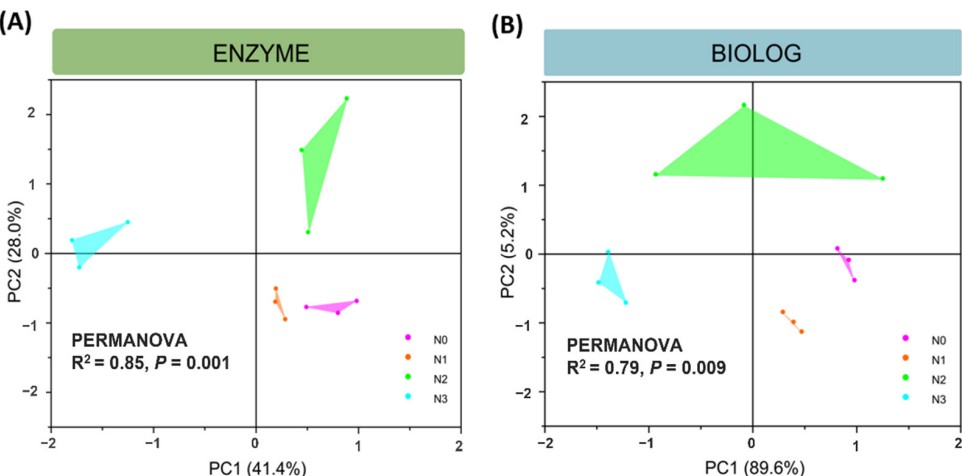

**Figure 2.** Principal component analysis (PCA) of Enzyme data (**A**) and Biolog profiles (**B**). Three replicates in the same color block represent a treatment. The *p* values between treatments refer to PERMANOVA results.

### 3.3. Microbial Functional Characteristics (CLPP)

After N addition, a negative effect on the diversity and richness of soil microbial community function was found in our study (Figure 3). Total AWCD, Shannon, and McIntosh indices decreased with the increase of N-addition rates, where a significant difference was found in the N3 (high-level) treatment. Similarly, the PCA result demonstrated that there was a distinct separation between N0, N1, and N3 treatments of total AWCD after 168 h

incubation (Figure 2B, PERMANOVA test, $p = 0.009$), although there was no significant difference between N1 and N2 treatments. In addition, polymers (0.30–1.41, range of AWCD in all treatments) were abundantly utilized by soil microorganisms, followed by amino acids (0.35–1.19), carboxylic acids (0.40–1.09), carbohydrates (0.17–1.00), and phenolic compounds (0.07–0.75), while amines (0.03–0.99) were rarely utilized. For these different primary C groups, the microbial utilization of amino acids, amines, carbohydrates, and polymers also showed a declining trend with N addition and had a significant effect on N3 treatment compared with the control except phenolic acids and carboxylic acids, ($p < 0.05$, Figure 3B–G). The phenolic acids utilized by soil microbes in N2 had a higher value than N1, mainly because of the highest level of D-glucosaminic acid and D-galacturonic acid. Likewise, an increase in 2-hydroxybenzoic acid utilization explained the rise of phenolic acid utilization in N2 treatment. It is worth noting that across 31 different carbon sources, only pyruvic acid methyl ester, D-xylose, and 2-hydroxybenzoic acid utilization after N input were higher than the control (Figure 4).

*3.4. Links among Soil Extracellular Enzyme Activities, Microbial C Sources Utilization, and Physicochemical Properties*

Pearson's correlation analysis explained the relationships between microbial function characteristics and extracellular enzyme activities in N-enriched soil (Figure 5A). The activities of N-acquisition enzymes and oxidases were significantly correlated with six group carbon sources after 2 years of N-addition experiments. Soil PPO and LAP activities showed a significantly positive correlation with the total AWCD, Shannon, and McIntosh indices. Similarly, soil PER activity was positively correlated ($p < 0.05$) with the total AWCD and McIntosh index. Conversely, the activity of soil NAG was negatively correlated ($p < 0.05$) with the total AWCD, Shannon, and McIntosh indices. Redundancy analysis (Figure 6) revealed that two N-acquisition enzymes LAP and NAG explained 59.7% and 20.1% of the variation in CLPP, respectively (Figure 6). The links between enzyme activities, C sources utilization, and edaphic properties were tested using the Mantel test (Figure 5B). We observed that the soil enzyme activities significantly correlated with pH, TN, and TP ($p < 0.05$), and the microbial C sources utilization was significantly correlated with pH, $NH_4^+$-N, $NO_3^-$N, TN, TP, and the C/N ratio.

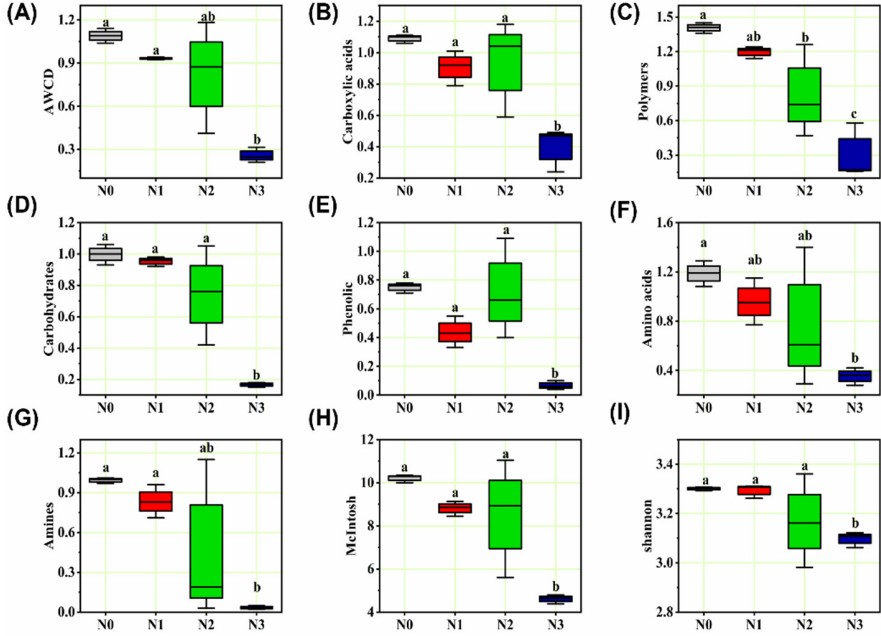

**Figure 3.** The average well-color development ((**A**), AWCD) of all C sources, different C sources groups (**B**–**G**), and diversity indices (**H**,**I**) among treatments in cucumber soil. Different lowercase letters on the boxplot indicate significant differences ($p < 0.05$) among treatments.

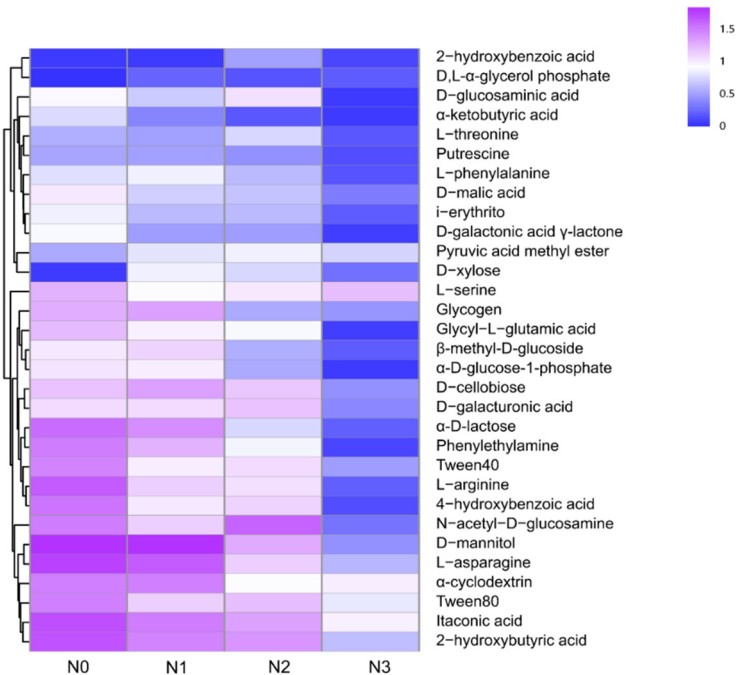

**Figure 4.** Heat map showing the utilization rates of 31 different Biolog carbon sources in N-amended (N0–N3) soil. The values are the mean for the time of mesocosms' incubation (168 h).

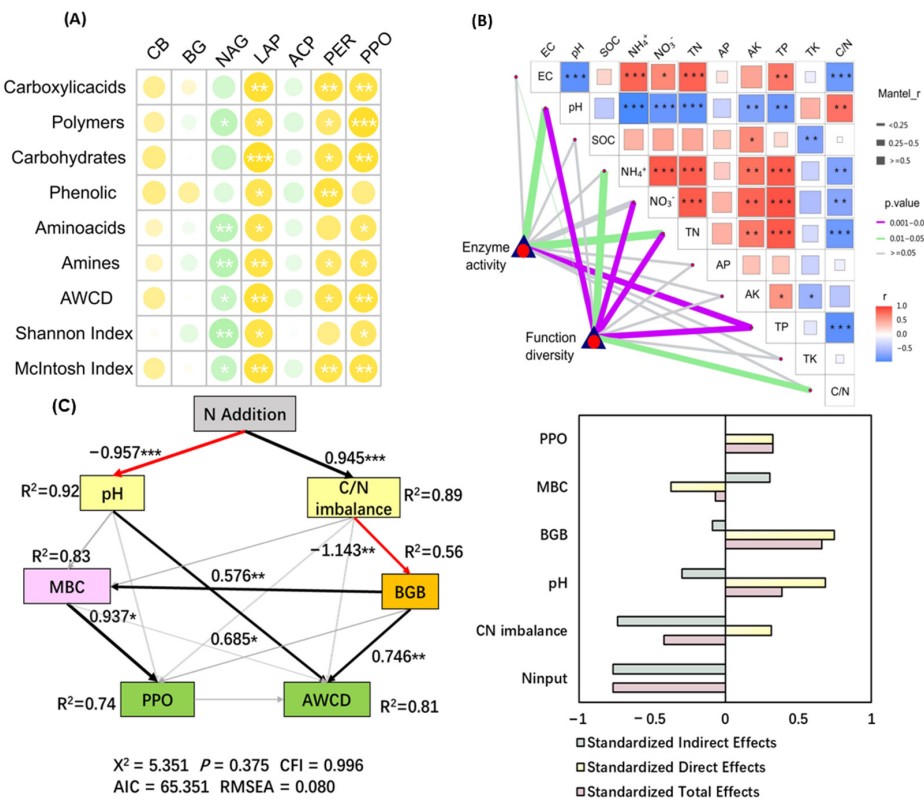

**Figure 5.** (**A**) Relationships between the enzyme-related parameters and the CLPP-related parameters. (**B**) Pearson correlations among edaphic and environmental factors. Microbial function and enzyme activities were linked to each factor by Mantel tests. Links width is proportional to Mantel' r, and link color signifies the level of significance. (**C**) The structural equation model (SEM) shows the potential causal relationships among soil-plant properties and microbial parameters. Data on the soil CN imbalance are the PC1 and PC2 results from the principal component analysis (PCA) of soil C (N)

(including soil TOC, TN, C/N, $NH_4^+$, $NO_3^-$). Black and red arrows represent significant positive and negative pathways, respectively. Arrow width corresponds to the path coefficient (numbers on the arrows). $R^2$ near the observed parameters denotes the proportion of the variance explained by other variables in the model. *** $p < 0.001$, ** $p < 0.01$, * $p < 0.05$.

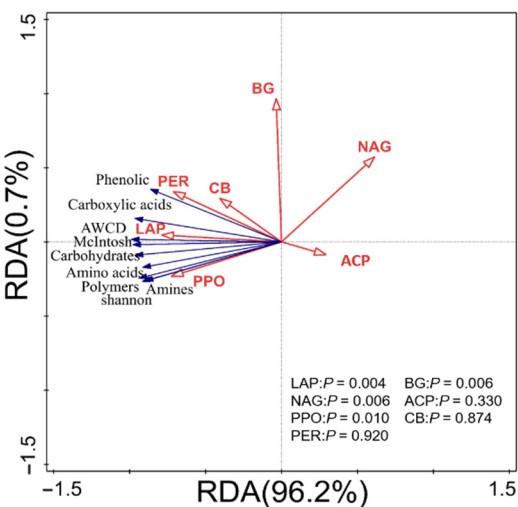

**Figure 6.** Redundancy analysis (RDA) between soil enzyme activities and microbial functional diversity.

## 4. Discussion

This study evaluated the effect of nitrogen addition on soil microbial function activities by altering soil and plant properties in the greenhouse. Consistent results between C-, N-, and P-cycle enzyme activities and microbial metabolic function (CLPP) in response to pH and C/N imbalance across four different N-fertilizer-addition rates indicated that the soil acidification and plant belowground biomass distribution might be the direct reason for the decrease in soil microbial functional diversity under an intensive cucumber cropping system.

As a vital regulation index, soil pH decreased by 0.63–1.52 units from the initial measurements, which was mainly caused by the release of protons during the oxidation of ammonium from the urea we applied as the nitrogen fertilizer [42]. Under N3 treatment, the soil EC reached 1.45 ms·cm$^{-1}$ (Table 2, $p < 0.05$). Higher ion concentration caused by excess N addition has severely restricted the growth of cucumbers and changed the composition of the rhizosphere microbial community [43]. In general, N-induced soil acidification leads to a loss of base cations, such as $K^+$, $Ca^{2+}$, $Na^+$, and $Mg^{2+}$ [44,45]. We noticed that the content of AK rose gradually with the increase of the N-addition rate, this might be related to the low molecular weight organic acids secreted by the roots, which increased the net negative charge on the surface of the variable charge soil, thereby increasing the absorption of potassium ions in the soil [46]. The content of the soil AP was significantly higher under N2 treatment (Table 2), suggesting that phosphorus limitation appeared in cucumber soil. Therefore, more available phosphatase was produced to supply plant absorption and utilization [47]. We further observed that the MBC content had a close linear relationship with cucumber root biomass and N availability (Figure S1), which implied that microbial activity was greatly impacted by soil nutrient level [9]. Under medium-high N-addition treatment (N2 and N3), the decrease of MBC is possibly due to the accumulation of toxic osmotic potential caused by soil acidification and secondary salinization [48].

Extracellular activities (involving carbon [C]-, nitrogen [N]- as well as phosphorus [P]-acquiring enzymes) are sensitive indicators for detecting modifications due to management practices [49]. Therefore, figuring out their response mechanism to N addition can help us better comprehend microbe nutrient demands and migration of soil elements. The results showed two N-acquisition enzymes (LAP and NAG) exhibited their respective response trends to N addition (Figure 1C,D). LAP activity decreased significantly with the increase in the N-addition rate. When soil nutrients were abundant, nitrogen input could reduce the

acquisition of N by microbes due to the decline of the DOC:DON ratio [50]. Furthermore, LAP was susceptible to environmental changes, and a strong positive correlation between the C/N ratio and LAP activity ($p < 0.05$) (Figure S1) further confirmed this view. In contrast, the activity of NAG remained higher under high nitrogen input (N3) treatment, which was well illustrated by the significantly negative correlation with soil pH (Figure S1; $p < 0.05$). NAG is an acid hydrolase which is involved in catalyzing the decomposition of chitin. Enhanced NAG activity following N addition might reflect factors other than N demand, such as fungal biomass, uptake of ions, and plant pathology [51,52]. For C-acquisition enzymes, BG and CB are related to the decomposition of labile C substrates such as carbohydrates, hemicellulose, and cellulose [53]. At N2 treatment, BG activity reached its highest ($p < 0.05$, Figure 1B). The emergence of mild carbon limitation induced by N availability might promote microbes investing more energy to acquire C sources. In forest and grassland ecosystems, Jing and Schleuss [21] found that enhanced BG activity is closely related to the increase in the abundance of C-degrading functional genes (endoglucanase, bglx, and bglb) under the addition of low and medium nitrogen. In addition, we also observed that cucumber belowground biomass of a low–medium N level was significantly higher than in high-N treatment. Under this suitable nutrient condition, BG and CB might be stimulated by root exudates as the substrate from the rhizosphere which in turn was beneficial to cucumber root growth and metabolism [54]. Unsimilar to the response of soil C- and N-cycle enzymes, the activity of ACP was significantly higher in all N-addition treatments and had a significant positive correlation with soil pH ($p < 0.05$, Figure S1). According to soil enzyme stoichiometry and vector analysis, we found that vector A values were less than 45 degrees, suggesting that acidification could induce microbial P limitation [55]. To meet the demand for phosphorus, microbes and plant roots devote more energy to producing acid phosphatase (Table 2 and Figure 1) [47,56]. Distinct from hydrolases, oxidases are mostly produced by fungi which are vital to the breakdown of resistant organic matter [57]. Oxidases (PPO and PER) activities were significantly inhibited under 1002 kg N ha$^{-1}$ year$^{-1}$ nitrogen addition (Figure 1). Structural equation modeling (SEM) results showed that PPO, was significantly influenced by MBC due to the C/N imbalance rather than the soil pH (Figure 5B). Compared with bacteria, fungi community composition and structure generally had a high tolerance for acidity, and the response to N availability appeared to be more intense [58]. Hence, compared to hydrolase, a relatively lower C/N ratio has been more likely to suppress the activities of oxidases such as PPO (Figure 5B), which might slow down the decomposition of SOC (Table 1; Figure S1) [59].

Our study confirmed that N addition had a significant effect on the CLPP of cucumber soil. The total AWCD and function diversity indices gradually decreased and reached a significant level under high N treatment (Figure 3), which was distinguished from previous studies [25,27,58]. For example, Zhang et al. [60] found increasing AWCD and Shannon index values at a low N-addition rate but decreasing when the rate was greater than 160 kg N ha$^{-1}$ year$^{-1}$. However, due to the high nutrient demand of vegetable crops in the greenhouse, the minimum addition rate we conducted was 334 kg N ha$^{-1}$ year$^{-1}$, which was much higher than other agroecosystems. Therefore, no upward trend in total AWCD and functional diversity indices were observed in all three N-addition treatments. According to the Mantel test, we found that pH and N availability (NH$_4^+$, NO$_3^-$) had a significant influence on microbial functional diversity (Figure 5B). There are two main reasons for this: (1) Wan et al. [61] reported rhizosphere bacteria community function and structure of greenhouses were significantly affected by the degree of soil acidification and identified a pH 5.5 as the limit value. In our experiment, the soil pH of N3 treatment was 5.37 which was classified as more acidic soil. The toxicity of hydrogen ions and aluminum ions caused by N-induced acidification inhibited the biomass and community composition of microorganisms, which in turn affected microbial functional metabolism [7,62]; (2) In addition, the input of high nitrogen led to a severe C/N imbalance in the soil (Table 2), which could restrict the underground distribution of plant photosynthetic products, and weaken the ability of microorganisms to acquire carbon sources [14]. The reduction of

MBC and total AWCD was confirmed by the reduction of root exudate and cucumber root biomass in our unpublished data.

We assessed how microbial function activities were influenced by N addition and its key drivers in N-enriched soils based on SEM model analyses (Figure 5C). The results showed that soil pH had a direct impact on microbial functional diversity and explained 38.8% of the variation in total AWCD. In contrast, a C/N imbalance indirectly decreased total AWCD by affecting cucumber belowground biomass. Meanwhile, the utilization rates of the other four C sources decreased gradually with the increase of the addition rate except for carboxylic acids and phenolic acid after two years of the field experiment. It is worth noting that N addition not only had a significant influence on the ability of soil microorganisms to utilize the six major C groups but also changed the consumption of single carbon sources [63,64]. The carbon source utilization gradually decreased with the increase of the N-addition rate. However, under low–medium N level, the utilization of D-Cellobiose, N-acetyl-D-glucosamine, D-Xylose, and pyruvic acid methyl ester by microorganisms remained higher in the cucumber soil compared to N0 (Figure 4). This could be explained by the activity of CB and NAG which had a relatively higher activity at a reasonable N input (Figure 1) and the accumulation of them was beneficial to the decomposition of D-Cellobiose and N-acetyl-D-glucosamine [65]. Therefore, microbes might prefer to use carbohydrates and carboxylic acids to balance the C/N ratio under low–medium situations [66]. Unlike N1 and N2 treatments, the microbial community in high N treatment (N3) might simultaneously suffer acid inhibition and strong carbon limitation [67]. Arshad and Ahmad [68] found the relative abundance of the Nitrospirae phylum and *Bacillus* genus involved in nitrogen and phosphorus cycling changed when the pH decreased significantly. Shifts in rhizosphere bacterial community composition might lead to changes in their function characteristics. Several specific N- or acid-sensitive bacteria and fungi competed with each other and exhibited a preference for some phenolic and amino acids such as 2-hydroxybenzoic acid and L-serine (Figure 4) from soil and root exudates to alleviate adverse environmental conditions although their AWCD values dropped [69]. It is worth noting that the yield of cucumber was positively correlated with the utilization of carbohydrates by the microbial community, which implied that certain microbial taxa associated with carbohydrate metabolism might play a vital role in evaluating the formation of cucumber yield ($p < 0.05$; Figure 7) [70]. However, the Biolog Eco-plates method only supports the analysis of the activity of the cultivatable microbial community [71] and it did not clearly reveal the regulatory mechanism of the microorganisms underlying the N-addition effect. Therefore, we need to investigate more methods to determine clear evidence in the future.

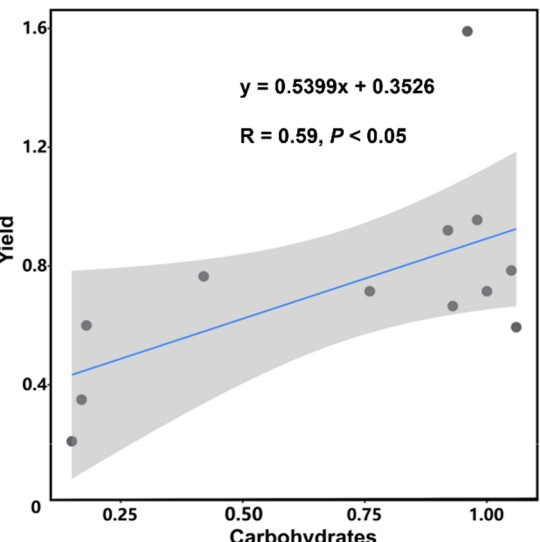

**Figure 7.** Correlation between the cucumber yield and carbohydrate utilization based on CLPP.

## 5. Conclusions

In this study, we demonstrated that excessive N addition decreased the microbial functional diversity through pH reduction and increased carbon limitation. Polymers, carbohydrates, and phenolic acid were sensitive carbon sources for the variation in microbial community-level physiological profiles. Furthermore, pH was the determinant driving changes in soil extracellular enzyme activities. Our results not only contributed towards improving the understanding of pathways that mediate soil microbial function activities but also provided theoretical guidance to farmers for sustainable production. Under greenhouse cultivation, low–medium level N ($334 \text{ kg N ha}^{-1} \text{ year}^{-1}$, $668 \text{ kg N ha}^{-1} \text{ year}^{-1}$) addition can be a good strategy to increase cucumber yield. However, to gain more insights into the plant–soil–microbe C-, N-, and P-cycle functions, further investigation of molecular biology approaches such as definite root metabolic pathways and microbial functional genes (as determined by metagenomic sequencing) with different N-addition rates is necessary.

**Supplementary Materials:** The following supporting information can be downloaded at: https://www.mdpi.com/article/10.3390/agriculture12091366/s1, Figure S1: Relationships between the environment-related parameters and the enzyme-related parameters. Figure S2: Relationships between the environment-related parameters and the CLPP-related parameters. Table S1: Basic properties of the tested soil.

**Author Contributions:** Conceptualization, Z.W.; methodology, Z.W. and T.B.; software, Z.W. and Y.L.; Validation, Z.W. and S.W.; formal analysis, Z.W.; investigation, Z.W. and G.W.; resources, H.F. and Z.S.; data curation, Z.W., Q.S. and M.A.; writing—original draft preparation, Z.W.; writing—review and editing, Z.W.; supervision, Z.S. All authors have read and agreed to the published version of the manuscript.

**Funding:** This study was financially supported by the Shenyang Science and Technology Project (21109308), China Agriculture Research System (CARS-23) and the National Natural Science Foundation of China (31902093).

**Institutional Review Board Statement:** Not applicable.

**Data Availability Statement:** Not applicable.

**Acknowledgments:** The authors would like to thank the students from the Shenyang Agricultural University for their assistance in the field experiments.

**Conflicts of Interest:** The authors declare no conflict of interest.

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
