# Peer review of "Effects of Nitrogen Addition on Soil Microbial Functional Diversity and Extracellular Enzyme Activities in Greenhouse Cucumber Cultivation"

_agriculture, doi:10.3390/agriculture12091366_

Round 1

Reviewer 1 Report

This study highlights the intensive work done by the authors and the topic is of interest. The results obtained are presented succinctly and comprehensibly.

Reviewer 2 Report

It is a good and well-written manuscript, but I have a minors comments:

1- Keywords: it will be better to use different words rather than Title words.

2- Introduction: need to present more information about the importance of Cucumber as an economical crop.

3- Soil extracellular enzyme activity assays: why did the authors determine those enzymes specifically??

4- Figs 1,2,3 need to be more clear for the readers.

Reviewer 3 Report

The current manuscript entitled “Effects of nitrogen addition on soil microbial functional diversity and extracellular enzyme activities in greenhouse cucumber cultivation” by Wang et al. evaluated the effects of four different rates of N addition on soil properties, extracellular enzyme activities, and community-level physiological profiles under 2-year greenhouse pot experiments. The study suggests that excessive N-fertilizer amendment can be more dangerous to microbial community function diversity, especially for carbohydrate utilization which adversely affects cucumber yield in current intensive management particularly in North China.  After a careful reading, I found this manuscript well-written, novel, and suitable for publication in the Agriculture journal. However, I would like to suggest some minor improvements which should be amended in the current version of the manuscript. My specific comments are:

1.      The abstract should indicate major numerical findings, thus, add maximum and minimum results values for respective parameters.

2.      L78: CLPP?

3.      L101: USDA? Full form.

4.      An experimental layout flowchart or table for all experimental treatments is desirable for better understanding.

5.      Be consistent while writing probability i.e., small ‘p’ or capital ‘P’.

6.      Some sentences regarding significant differences in different treatment groups need to be properly written in a more scientifically sound.

7.      Define all abbreviations under the table footnotes.

8.      Biolog carbon? Write completely.

9.      Figure 7: Better if you write the linear fitness equation (y=ax+b) also.

10.   References: fine.
